# CATBench: A Compiler Autotuning Benchmarking Suite for Black-box Optimization

**Jacob O. Tørring**[1,*]   **Carl Hvarfner**[2,*]   **Luigi Nardi**[2]   **Magnus Själander**[1]

[*]Equal contribution.
[1]Norwegian University of Science and Technology (NTNU)
[2]Lund University

**Abstract**   Bayesian optimization is a powerful method for automating tuning of compilers. The complex landscape of autotuning provides a myriad of rarely considered structural challenges for black-box optimizers, and the lack of standardized benchmarks has limited the study of Bayesian optimization within the domain. To address this, we present CATBench, a comprehensive benchmarking suite that captures the complexities of compiler autotuning, ranging from discrete, conditional, and permutation parameter types to known and unknown binary constraints, as well as both multi-fidelity and multi-objective evaluations. The benchmarks in CATBench span a range of machine learning-oriented computations, from tensor algebra to image processing and clustering, and use state-of-the-art compilers, such as TACO and RISE/ELEVATE. CATBench offers a unified interface for evaluating Bayesian optimization algorithms, promoting reproducibility and innovation through an easy-to-use, fully containerized setup of both surrogate and real-world compiler optimization tasks. We validate CATBench on several state-of-the-art algorithms, revealing their strengths and weaknesses and demonstrating the suite's potential for advancing both Bayesian optimization and compiler autotuning research.

## 1 Introduction

Bayesian optimization (BO) [37, 31, 47, 18, 20] is a powerful tool for automating the optimization of resource-intensive black-box systems, such as machine learning hyperparameter optimization [16, 44, 28, 29, 30], hardware design [38, 15], and scientific discovery [24, 36, 50]. By intelligently exploring the configuration space and learning from observed performance, BO can efficiently identify high-performing configurations with limited resource consumption.

*Autotuning* [1, 5, 54, 55, 58, 4, 35] is the black-box optimization process of any performance-impacting parameters in a general program on a hardware platform. The impact of optimization is profound, and efficiency gains of up to 1.5x - 11.9x are frequently observed in practice [4, 56, 26]. However, autotuning is not without its challenges: the search spaces are commonly discrete, categorical, and permutation-based and may involve both known and unknown constraints on the parameter space. Moreover, multiple output metrics are of relevance [45], and performance differs across hardware and the type of computation that is to be performed [56, 54, 4]. As such, despite the importance of autotuning for the performance of compiled programs, most applications of BO in this domain [59, 54] have not been adapted to address these unique challenges. In-depth design of BO methods for this domain has thus been limited, with the notable exceptions of the Bayesian Compiler Optimization framework (BaCO) [26] and GPTune [35, 4].

To facilitate further research and development in BO for autotuning, it is crucial to have a diverse and representative set of benchmarks. A comprehensive benchmarking suite is necessary to evaluate optimization algorithms effectively, ensuring that they can handle the complex and varied nature of real-world autotuning problems. We introduce CATBench[1], a comprehensive benchmarking

---

[1]https://github.com/odgaard/catbench

suite designed to evaluate black-box optimization algorithms in the context of compiler autotuning. CATBench builds upon the real-world applications used in the BaCO [26] framework, incorporating additional parameters, fidelity levels, and multiple objectives. The suite encompasses a wide range of domains, including tensor algebra, image processing, and machine learning, and targets various backend hardware platforms, e.g., Intel CPUs and Nvidia GPUs. By providing a standardized set of real-world benchmarks with a unified, containerized interface usable on a variety of hardware, CATBench enables fair benchmarking of optimization algorithms and facilitates reproducibility. Additionally, the architecture of our benchmarking suite makes it easy to extend the benchmark suite with new compiler frameworks and autotuning problems.

The main contributions of this paper are as follows:

1. A comprehensive benchmark suite consisting of ten real-world compiler optimization tasks, which encompass mixed discrete, categorical, and permutation-based search spaces, incorporate both known and unknown binary constraints, and support multiple tuning objectives and multi-fidelity information sources.
2. A benchmarking framework that provides a simple interface and makes it easy to prototype using surrogate models and run large-scale experiments on clusters of server machines,
3. Thorough evaluation of popular BO methods and Evolutionary Algorithms showcasing the properties of the benchmarking suite. Since the existing optimization methods do not cover all the features needed by autotuning, we include necessary adaptations for compatibility with the requirements of CATBench.

## 2 Background

### 2.1 Bayesian Optimization

BO aims to find a minimizer $x^* \in \arg\min_{x \in \mathcal{S}} f(x)$ of the black-box function $f(x) : \mathcal{S} \to \mathbb{R}$, over some possibly discontinuous, non-euclidean input space $\mathcal{S}$. $f$ can only be observed pointwise and its observations are perturbed by Gaussian noise, $y(x) = f(x) + \varepsilon_i$, where $\varepsilon_i \sim \mathcal{N}(0, \sigma_\varepsilon^2)$. The *Gaussian processes* (GPs) is the model class of choice in most BO applications. A GP provides a distribution over functions $\hat{f} \sim \mathcal{GP}(m(\cdot), k(\cdot, \cdot))$, fully defined by the mean function $m(\cdot)$ and the covariance function $k(\cdot, \cdot)$. Under this distribution, the value of the function $\hat{f}(x)$ at a given location $x$ is normally distributed with a closed-form solution for the mean $\mu(x)$ and variance $\sigma^2(x)$. We model a constant mean, so the covariance function fully determines the dynamics $k(\cdot, \cdot)$. The *acquisition function* uses the surrogate model to quantify the utility of a point in $\mathcal{S}$. Acquisition functions balance exploration and exploitation, typically employing a greedy heuristic. The most common is Expected Improvement (EI) [31, 3], but other acquisition functions, such as Thompson sampling [53] and Upper Confidence Bound [49, 48], are also frequently utilized [17, 57].

### 2.2 Compiler Optimization

Compiler autotuning is the process of automatically optimizing compiler parameters to enhance the performance of software on specific hardware backends. The task at hand is to generate code that minimize objectives such as program execution time, memory use, storage size, and power consumption for a given low-level program, such as matrix-matrix multiplication [26]. Tunable parameters include loop ordering, the number of threads for parallel execution, data chunk sizes, and various boolean flags related to inlining and loop unrolling, among multiple others. These parameters enable us to optimize the memory access patterns so that more of the program data can stay in the CPU cache, which in turn accelerates the program to complete faster.

This optimization process is crucial for achieving efficient execution of programs, particularly on modern heterogeneous computing architectures, such as central processing units (CPUs), graphical processing units (GPUs), and field-programmable gate arrays (FPGAs). Autotuning typically

leverages heuristic search algorithms [1], evolutionary methods [9, 25], or BO [26, 60, 61] to explore the vast configuration spaces, which include discrete, categorical, and permutation-based parameters, while considering both known and unknown constraints.

## 2.3 Related Work

**Bayesian Optimization over Discrete Search Spaces**. BO has seen multiple adaptations to accommodate alternatively structured search spaces. SMAC [27, 34] natively supports discrete and categorical parameter types due to its use of a Random Forests surrogate. [39, 11, 41, 43, 57] propose kernels that handle integer-valued or categorical parameters, while [8, 21] focus on optimizing the acquisition function in this context. Additionally, [40, 10] consider BO algorithms specifically designed for permutation-based search spaces. The BaCO framework [26] integrates multiple components to address the unique characteristics of compiler search spaces. This includes categorical [43, 57] and permutation-based [10] kernels, acquisition functions for handling unknown constraints [19, 22], and Random Forests surrogate models for constraint prediction [27].

**Autotuning Methods**. In the context of compiler autotuning, BO has been successfully applied to find high-performing compiler configurations for various programming languages, compilers, and target architectures [2, 4, 59, 54, 26]. The autotuning problem is formulated as a black-box optimization problem, where the objective is the performance metric of interest (execution, energy usage), and the input space consists of compiler flags, parameters, and transformations.

**Benchmarking Suites**. Autotuning lacks a standardized benchmarking suite, despite various efforts to develop such resources [42, 23, 52, 56, 60]. In Table 1, we present a comparative analysis of our benchmarking suite against these prior works. PolyBench [23, 60] offers a substantial collection of benchmarks, each with multiple input datasets. While some benchmarks include up to ten optimization parameters, they are predominantly boolean parameters (which is a specific instance of categorical parameters), with none being permutation variables, resulting in unrealistically small search spaces that are not representative of real-world autotuning applications. Furthermore, PolyBench does not emphasize multi-objective optimization or multi-fidelity parameters and consists of C-based benchmarks without a Python interface. KTT [42] exhibits similar limitations to PolyBench, although it provides benchmarks with larger search spaces. BAT [52, 56] features a suite of Nvidia CUDA GPU kernels, accessible via a Python interface, with generally larger search spaces than those in PolyBench and KTT. Nevertheless, each benchmark is limited to a single input dataset and lacks permutation variables, unknown constraints, and support for multi-objective tuning or multi-fidelity information sources. In the category of reinforcement-learning agents there is also the notable work of CompilerGym [7], an extensive suite of compiler environments.

In the context of BO, a large collection of benchmarking suites has been proposed. primarily in the context of Hyperparameter Optimization (HPO) [65, 33]. HPOBench [14] provides a collection of models for evaluating hyperparameter optimization algorithms, and captures a range of parameter types as well as multi-fidelity evaluations, whereas [62, 12, 64] target benchmarking of Neural Architecture Search (NAS) algorithms. MCBO [13] collects discrete and categorical tasks and BO algorithms in a modular structure. Lastly, BaCO [26] introduced a set of benchmarks for evaluating BO algorithms in the context of autotuning, derived from real-world applications and compilers, covering a range of domains and target architectures.

CATBench is a comprehensive suite based on BaCO, which: 1) expands the Tensor Algebra COmpiler (TACO) benchmarks with additional parameters and options, 2) implements and exposes multiple multi-fidelity and multi-objective parameters, and 3) incorporates CPU and GPU energy measurements to extend the benchmarks to multi-objective problems.

| Name | Types | Bench. | $M$ | $D$ | Datasets | $F$ | $|\mathcal{S}|$ | Cons. | Python |
|------|-------|--------|-----|-----|----------|-----|-----------------|-------|--------|
| PolyBench | OC | 30 | 1 | $3-10$ | 5 | 0 | $10^2 - 10^5$ | K | N |
| KTTBench | OC | 10 | 1 | $5-15$ | 1 | 0 | $10^2 - 10^5$ | K | N |
| BAT | OC | 8 | 1 | $6-10$ | 1 | 0 | $10^2 - 10^6$ | K | Y |
| **CATBench** | OCP | 10 | $2-3$ | $4-10$ | $1-15$ | $2-4$ | $10^3 - 10^8$ | K/H | Y |

Table 1: Benchmark characteristics for various benchmark suites. In order, left to right: Name of the benchmark suite, exposed parameter types (O=ordinal, C=categorical, P=permutations), number of benchmarks *Bench.*, number of exposed objectives $M$, input dimensionality $D$, number of input datasets, number of fidelity dimensions $F$, search space cardinality $\mathcal{S}$, the types of constraints included (K=known, H=hidden), and whether or not the benchmarks are easily callable from Python.

## 3 The CATBench Benchmarking Suite

CATBench is a comprehensive benchmarking suite designed to evaluate BO algorithms in the context of autotuning, which presents unique challenges to black-box optimization. These challenges include a unique combination of conditions such as permutation variables, known and unknown constraints, and variations in search spaces between hardware architectures [26]. CATBench comprises a diverse set of benchmarks derived from real-world applications and compilers, covering various domains, hardware platforms, and optimization challenges. The suite is accessible through an intuitive Python interface, enabling users to quickly prototype novel algorithms with surrogate models or deploy the benchmarks to a cluster of servers using our client-server architecture.

### 3.1 The CATBench Interface

The CATBench Python interface facilitates benchmarking via a client-server architecture and Docker containers. The interface significantly simplifies the evaluation process by providing a consistent environment for performance assessments. The code snippet illustrates a typical benchmarking workflow using CATBench, using either a surrogate model or a hardware benchmarking server:

```python
import catbench as cb

study = cb.benchmark("asum") # Surrogate example
study = cb.benchmark("asum", dataset="server", # Hardware example
    server_addresses=["your_benchmark_server"])

space = study.definition.search_space
optimizer = your_optimizer_setup(space)

while optimizer.not_done:
    config, fidelities = optimizer.next()
    # e.g. fidelities: { "iterations": 10 }

    result = study.query(config, fidelities)
    optimizer.update(result)
```

Listing 1: Benchmarking with CATBench

A naive approach could involve an interface that directly calls the TACO and RISE benchmarks using Python subprocesses. However, this method is inefficient due to the high overhead of repeatedly starting and initializing a benchmark for each configuration, including reloading all input datasets, making the process extremely time-consuming. To overcome this inefficiency, we designed a client-server benchmark setup. By starting the server once, we can initially load the program and datasets, then continuously listen for and execute new configurations, significantly reducing initialization overhead and streamlining the benchmarking process.

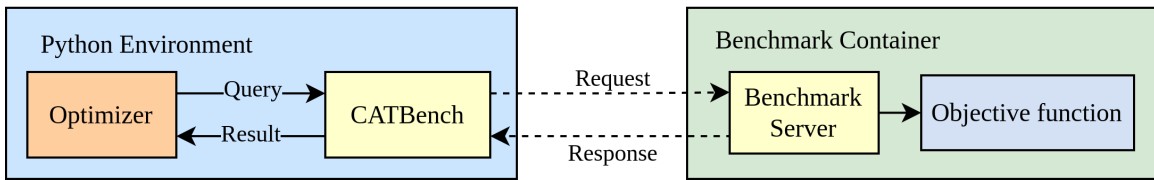

Figure 1: The CATBench network-based client-server model.

The architecture, depicted in Figure 1, builds on the client-server interface concept introduced by HyperMapper [38] and BaCO [26], extending it to a network-oriented protocol. We adopt Google's RPC framework, *gRPC*, which is well-suited for network communication. With gRPC, benchmarks can be implemented in any programming language while maintaining seamless communication with a Python-based optimizer. This decoupling enables deploying benchmarks on separate machines and optimizing them from another machine, as shown in Figure 1. Consequently, this setup allows for creating a scalable server cluster to run benchmarks and distribute tasks across machines.

CATBench provides surrogate models of the benchmarks for quick prototyping and piloting. These surrogate models enable preliminary testing and optimization without requiring specific GPUs or CPUs, and can soften particular requirements of the real benchmark. The surrogates aim to provide value in early stages of development, when computational resources may be limited, or quick iterations are needed to inform decisions on algorithm design. We employ Docker containers to encapsulate benchmarking tools and their dependencies, ensuring consistency and reproducibility across diverse environments. This approach enables the user to easily set up an environment that mitigates software version conflicts and system incompatibilities, ensuring reliable and comparable results. This approach, akin to NASBench [63] in the context of NAS, is outlined in App. A.

## 3.2 Benchmark Characteristics

Table 2 provides a detailed overview of the benchmarks included in CATBench. The table presents key characteristics and statistics of each benchmark, allowing for a comprehensive understanding of their complexity and optimization challenges. The benchmarks are categorized into two groups based on the types of hardware, tasks, and the compiler used: The TACO tasks, which are based on the Tensor Algebra Compiler (TACO) [32] and address the optimization of computations made on the CPU, and RISE/ELEVATE [51], which are based on the RISE data parallel language and its optimization strategy language ELEVATE and address computations made on the GPU. Apart from the types of programs in each group of benchmarks, the groups of benchmarks additionally differ in their search spaces, number of objectives, fidelities, and constraint characteristics.

### 3.2.1 TACO.
The TACO benchmarks focus on sparse matrix, vector, and tensor computations on the CPU. The benchmarks include sparse matrix multiplication (SpMM), sparse matrix-vector multiplication (SpMV), sampled dense-dense matrix multiplication (SDDMM), tensor times vector (TTV), and matricized tensor times Khatri-Rao product (MTTKRP). The search spaces involve ordinal, categorical, and permutation parameters, with known and unknown constraints on how the program can be compiled. The objective is to minimize the execution time and the CPU energy consumption of the generated code. TACO is a compiler written in C++ that generates, compiles, and runs a C program to solve the tensor algebra expression based on the input configuration provided by the user. The optimal choice of values for these optimization parameters will vary based on the specific hardware architecture details of the CPU that the program is running on. See subsection B.1 for more details on the TACO benchmarks.

The TACO benchmarks are characterized by a mix of ordinal, categorical, and permutation parameters. For example, the loop ordering parameter in these benchmarks is a permutation parameter that determines the order in which nested loops are executed—a critical factor affecting

| Group | Benchmark | Params | $M$ | $D$ | $F$ | $\|\mathcal{V} \subseteq \mathcal{S}\|$ | $\frac{\|\mathcal{V}\|}{\|\mathcal{S}\|}$ | Cons. | Hardware |
|---|---|---|---|---|---|---|---|---|---|
| **RISE** | GEMM | O | 3 | 10 | 4 | $156 \cdot 10^6$ | 1.34% | K/H | GPU |
| | Asum | O | 3 | 5 | 4 | $61.7 \cdot 10^3$ | 4.80% | K | GPU |
| | Kmeans | O | 3 | 4 | 4 | $3.62 \cdot 10^3$ | 24.70% | K/H | GPU |
| | Scal | O | 3 | 7 | 4 | $4.24 \cdot 10^6$ | 10.70% | K/H | GPU |
| | Stencil | O | 3 | 4 | 4 | $3.64 \cdot 10^3$ | 24.90% | K | GPU |
| **TACO** | SpMM | OCP | 2 | 8 | 2 | $310 \cdot 10^3$ | 5.40% | K | CPU |
| | SpMV | OCP | 2 | 9 | 2 | $14.2 \cdot 10^6$ | 10.69% | K | CPU |
| | SDDMM | OCP | 2 | 8 | 2 | $576 \cdot 10^6$ | 3.71% | K | CPU |
| | TTV | OCP | 2 | 9 | 2 | $17.8 \cdot 10^6$ | 21.20% | K/H | CPU |
| | MTTKRP | OCP | 2 | 8 | 2 | $5.87 \cdot 10^6$ | 19.70% | K | CPU |

Table 2: Properties of each benchmark in CATBench. In order, left to right: Exposed parameter types (O=ordinal, C=categorical, P=permutations), number of exposed objectives $M$, input dimensionality $D$, number of fidelity dimensions $F$, number of known valid configurations $|\mathcal{V} \subseteq \mathcal{S}|$, ratio of known valid configurations $\frac{|\mathcal{V}|}{|\mathcal{S}|}$, and the types of constraints (K=known, H=hidden).

memory access patterns and cache utilization. The benchmarks have relatively large search space sizes and a high percentage of valid configurations. The TACO benchmarks also feature known constraints. Compared with the TACO benchmarks from BaCO [26] we have extended the search space with two new parameters, added four multi-fidelity parameters, and added CPU energy as an additional objective.

**3.2.2 RISE/ELEVATE.** The RISE/ELEVATE benchmarks are derived from the RISE/ELEVATE [51] compiler framework. The RISE programming language and ELEVATE configuration language specify how the framework should generate an optimized GPU OpenCL kernel. The benchmarks cover dense linear algebra (GEMM, Asum, Scal), clustering (Kmeans), and stencil computations (Stencil). The objectives are to minimize execution time of the generated code on the GPU, while minimizing energy on the CPU, the GPU, or both. See subsection B.2 for details of the benchmark computations. The benchmarks involve ordinal parameters and have a relatively low percentage of valid configurations due to substantial known and hidden constraints. The search spaces involve a mix of ordinal and continuous parameters, with known as well as hidden constraints, which are exposed when a configuration is queried. These benchmarks have moderate search space sizes and multiple fidelity dimensions.

**3.2.3 Constraint Handling.** We expect the optimization algorithms to handle known constraints directly, which can avoid querying invalid configurations. Unknown (hidden) constraints, which are only discovered upon evaluation, are modeled by returning a constant poor objective value when a configuration violates such constraints. Invalid configurations are not discarded but rather assigned penalty values to maintain a consistent optimization interface.

**3.2.4 Multi-fidelity and Multi-objective Benchmarks.**

**Multi-fidelity Evaluation** As the evaluations of both energy usage and runtime of compiled programs are typically subject to substantial noise, it is conventional to evaluate a configuration tens of times and use the average (median or mean) observation for optimization purposes. Moreover, there is an additional option to repeat runs several times, where the cache is cleared before re-evaluation to eliminate a source of bias. After evaluation, transitory background noise might appear from periodic background disturbances. To counteract these events, we allow the user to specify the wait time between benchmark repeats. Finally, we also allow for the control of the wait period after the benchmarking is finished before a new configuration may start. This process exposes four fidelity parameters: the number of iterations to compute the kernel, the re-evaluations after the

| Benchmark | CPU/GPU | #Cores | Memory | Figure Labels |
|-----------|---------|--------|--------|---------------|
| TACO | 2x Intel Xeon E5-2630 v4 | $2 \times 10$ | 128 GB DDR4 | XeonE5 |
| | 2x Intel Xeon Gold 6242 | $2 \times 16$ | 768 GB DDR4 | XeonG |
| | 2x AMD Epyc 7742 | $2 \times 64$ | 256 GB DDR4 | Epyc |
| RISE/ELEVATE | Nvidia Titan RTX | 4608 | 24 GB GDDR6 | TitanV / TV |
| | Nvidia Titan V | 5120 | 12 GB HBM2 | RTXTitan / RTX |

Table 3: Hardware Specifications for TACO and RISE/ELEVATE Benchmarks.

cache is cleared, wait between repeats, and a wait after evaluation. All benchmarks in CATBench support multi-fidelity evaluation along multiple dimensions, to facilitate complex cost-efficient optimization. The number of fidelity dimensions is displayed in the *F* column of Table 2. In our experiments, we used default fidelity settings of 10 iterations for kernel execution, 5 repeats with cache clearing between each repeat, 1 second wait between repeats, and 10 seconds wait between configurations. While we did not perform extensive evaluation of multi-fidelity optimization algorithms in this work, these parameters provide a foundation for future multi-fidelity studies. For TACO benchmarks specifically, the iterations parameter controls how many times the kernel is called in the inner loop, while repeats determines the number of outer loop executions with cache clearing between each repetition.

**Multi-Objective Evaluations** All benchmarks in CATBench are amenable to multiple optimization objectives, where the objectives display varying degrees of conflict, as visualized in Figure 6. The exposed objectives are program runtime and energy usage. The "Obj" column in Table 2 indicates the number of objectives for each benchmark.

### 3.2.5 Variability Across Hardware.

The optimal compiler configuration varies with different hardware, making each hardware effectively a new CATBench benchmark. As shown in Figure 4, we illustrate this by running a set of configurations on three tasks, SpMV, SDDMM, and Stencil, and plotting the distribution of output values. The output values' distribution varies between tasks and indicates that the tasks are similar but not identical. This demonstrates CATBench's utility in a transfer learning context: practitioners can optimize tasks on one hardware and use transfer learning to warm-start optimization on other models.

Benchmark variability necessitates running all algorithms on the same hardware during an optimization round to ensure consistency. Additionally, CATBench benchmarks are influenced by hardware conditions, such as operational temperature, affecting performance. Significant evaluations should be conducted on managed compute clusters to minimize external factors. Since performance evaluations on real hardware are inherently noisy and environment-dependent, CATBench provides surrogate tasks to model the underlying objective for reproducibility in controlled settings.

## 4 Results

We now assess various properties of CATBench. We demonstrate the performance of various optimization algorithms on the CATBench, in a single-objective context on TACO, and for multiple objectives on RISE/ELEVATE. We evaluate three optimization algorithms: BaCO, NSGA-II, and Random Search. To provide a diverse display, the TACO benchmarks focus on single-objective optimization for compute time, whereas for the RISE/ELEVATE benchmarks, we perform multi-objective optimization considering both compute time and GPU energy consumption. We assess variability across hardware for three tasks and analyze feature importance for the Stencil and SPMM benchmark. Lastly, we provide insight on multi-objective components of CATBench. Table 3 outlines hardware specifications used for our experiments, which determines benchmark characteristics.

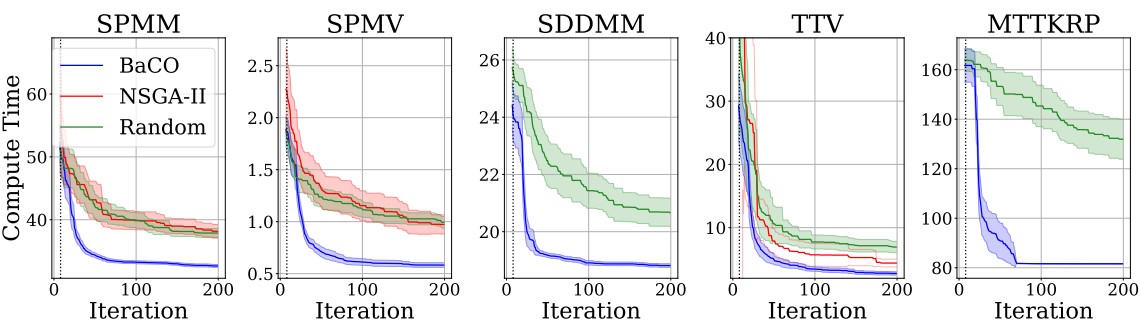

Figure 2: Minimum average compute time including two standard errors per optimization algorithm on the TACO optimization tasks on XeonE5. BaCO substantially outperforms non-BO algorithms. NSGA-2 encountered numerical errors on SDDMM and MTTKRP.

## 4.1 Optimization Results

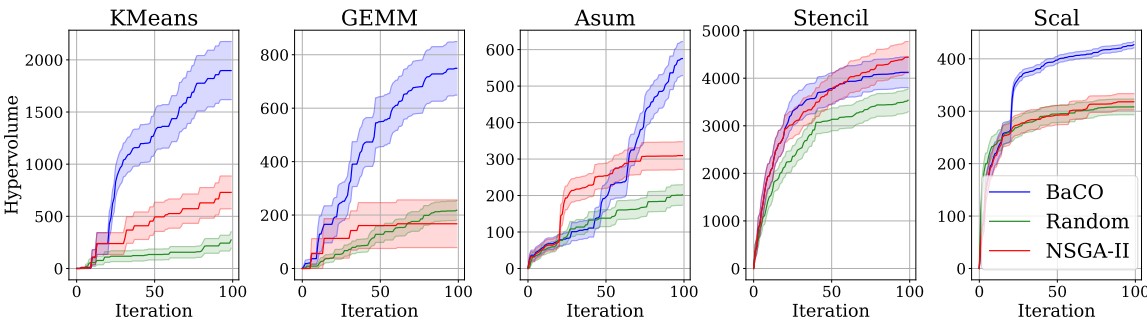

Figure 3: Hypervolume improvement on compute time and GPU energy consumption from a reference point, including two standard errors per optimization algorithm on the RISE/ELEVATE optimization tasks on RTXTitan. BaCO substantially outperforms non-BO algorithms.

We evaluate the optimization performance of three algorithms, BaCO, NSGA-II, and Random Search, on `CATBench`. For the TACO results in Figure 2, we run single-objective optimization on the compute time of the kernels. For our RISE/ELEVATE benchmarks, we run multi-objective optimization with compute time and GPU energy consumption as seen in Figure 3. Here, BaCO significantly outperforms the non-BO algorithms, with NSGA-II encountering numerical errors on SDDMM and MTTKRP. For the RISE/ELEVATE benchmarks, we perform multi-objective optimization considering both compute time and GPU energy consumption. Figure 6 illustrates that BaCO also substantially outperforms the other algorithms in this context, in all tasks but Stencil.

## 4.2 Understanding the `CATBench` Benchmarks

**Variability Across Hardware** We evaluate the variability across hardware for three tasks to assess the general variation, and assess the potential for `CATBench` in the context of meta-learning. In Figure 4, we plot the empirical PDF for the computational speedup of 5000 random samples across three tasks (SpMV, SDDMM, and Stencil) and different hardware types, specified in Table 3. Depending on hardware, tuning these applications can deliver between a 10x improvement in performance from a non-tuned, default setting application.

**Feature Importance** Many of the `CATBench` compilers have similar search spaces, but their characteristics may vary substantially. In Figure 5, the permutation feature importance for both objectives on the Stencil benchmark run on a TitanRTX (RTX) and a TitanV (TV), and the same metric for compute time of the SPMM benchmark across hardware. Some parameters, like *omp_-*

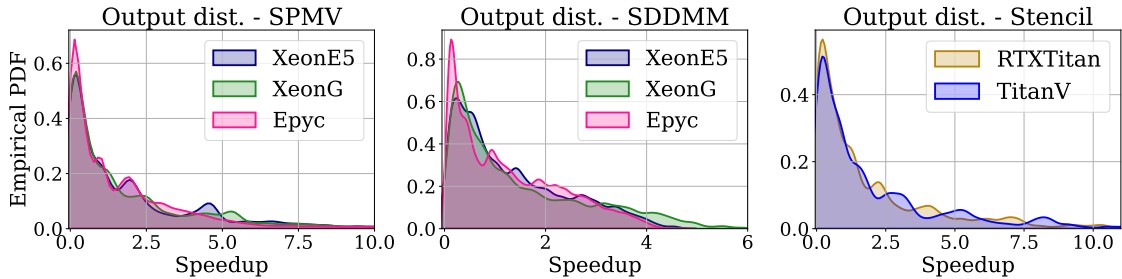

Figure 4: Empirical density of speedups for SPMV, SDDMM, and Stencil for three anthreed two sets of hardware, respectively. The distribution of output is similar across tasks, suggesting that a change in hardware yields a similar, albeit not identical, task.

*num_threads* for SPMM, and *tuned_gs0* and *tuned_gs1* for Stencil, have substantially varying feature importance. The benchmarks are moderately sparse in the number of high-impact dimensions.

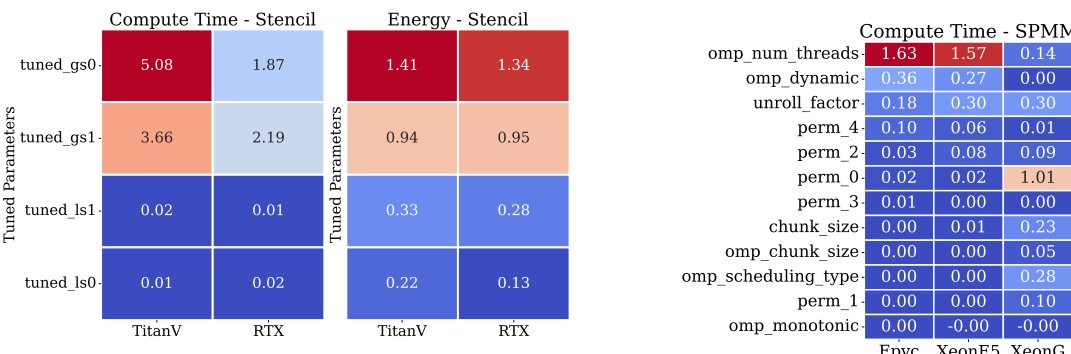

Figure 5: (left) Feature importance for both objectives on the Stencil benchmark run on a TitanRTX (RTX) and a TitanV (TV). (right) Feature importance for compute time for the SPMM benchmark across hardware. The *omp_num_threads* parameter is the most important on both Epyc and XeonE5, while only marginally impactful when run on XeonG. While all objectives are fairly sparse, the feature importances can vary substantially between hardware.

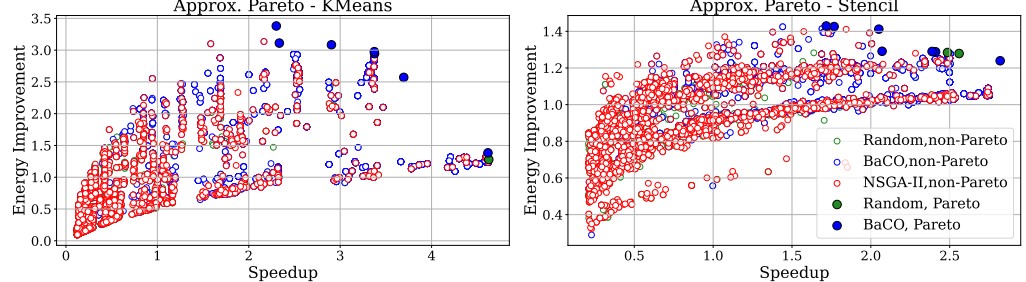

Figure 6: Computational speedup and energy efficiency for all explored configurations on the Kmeans and Stencil benchmarks. The objectives display moderate correlation, and Pareto front configurations exhibit various trade-offs in computational speed-up.

**Multi-Objective Trade-offs**. Figure 6 highlights the multi-objective trade-offs that CATBench exposes by displaying approximate trade-offs between the two objectives, runtime and energy consumption, for the two RISE tasks KMeans and Stencil. The two objectives are moderately correlated for lower-performing configurations, yet produce a broad Pareto front with diverse characteristics.

## 5 Conclusion

We introduce CATBench, a benchmarking suite for evaluating BO algorithms in compiler autotuning. CATBench features a diverse set of benchmarks derived from real-world applications, covering various domains, hardware platforms, and optimization challenges. These benchmarks are selected to reflect the unique characteristics of compiler autotuning tasks. CATBench offers a novel set of challenging tasks to BO through exotic search spaces, multi-objective and multi-fidelity evaluations, and transfer learning capabilities. By providing a standardized set of benchmarks and a unified evaluation interface, CATBench enables fair comparison reproducibility, while enabling accelerated progress towards more efficient and effective autotuning methods.

CATBench aims to be a valuable resource for researchers and practitioners, enabling the development and evaluation of novel algorithms, identifying strengths and weaknesses of existing methods, and assessing the impact of different design choices. Future work includes expanding CATBench with additional benchmarks from emerging domains, incorporating new performance metrics, and establishing a public leaderboard and repository of state-of-the-art results akin to DAWNBench [6].

## Acknowledgements

Luigi Nardi was supported in part by affiliate members and other supporters of the Stanford DAWN project — Ant Financial, Facebook, Google, Intel, Microsoft, NEC, SAP, Teradata, and VMware. Carl Hvarfner and Luigi Nardi were partially supported by the Wallenberg AI, Autonomous Systems and Software Program (WASP) funded by the Knut and Alice Wallenberg Foundation. Luigi Nardi was partially supported by the Wallenberg Launch Pad (WALP) grant Dnr 2021.0348. The experiments were performed on resources provided by Sigma2 — the National Infrastructure for High-Performance Computing and Data Storage in Norway, as well as the IDUN [46] computing cluster at NTNU.

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

## A  Surrogate Models

For a subset of benchmarks, we provide surrogate models to aid in algorithm development. The surrogates are CatBoost models that are trained on a large dataset, accumulated throughout our experimentation phase of the benchmarks. In Fig. 7, we show the R2-scores as a measure of prediction quality for all surrogates on the included tasks. Prediction errors are generally low, with the primary exceptions being TTV and MTTKRP. On Stencil, predictions are perfect, which can be attributed to the small search space. Notably, the surrogates do not enforce neither hidden nor known constraints, as they can predict the values of otherwise infeasible configurations. As such, the surrogate tasks are practical for developing an algorithm, one component at a time.

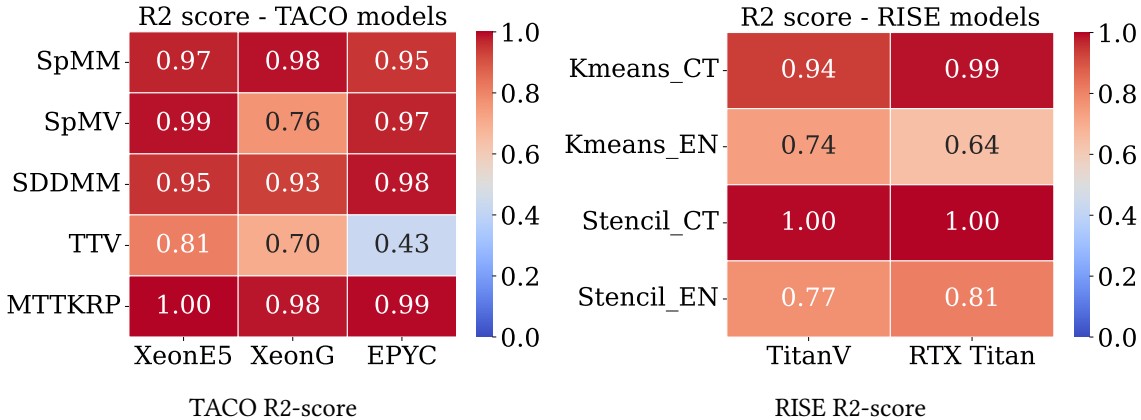

Figure 7: R2 scores on all tasks for which CatBoost surrogates are provided. Scores are generally high, with notable exceptions in TTV and MTTKRP on specific hardware.

## B  Benchmark Details

### B.1  TACO

The TACO benchmarks are based on the Tensor Algebra Compiler (TACO) [32] and focus on sparse tensor computations. The benchmarks include sparse matrix multiplication (SpMM), sparse matrix-vector multiplication (SpMV), sampled dense-dense matrix multiplication (SDDMM), tensor times vector (TTV), and matricized tensor times Khatri-Rao product (MTTKRP). The search spaces involve ordinal, categorical, and permutation parameters, with known constraints. The objective is to minimize the execution time of the generated code on CPUs.

The TACO benchmarks in CATBench focus on sparse tensor computations and are derived from TACO [32]. These benchmarks include:

**Sparse Matrix Multiplication (SpMM):** The SpMM operation computes the product of a sparse matrix $A$ and a dense matrix $B$, resulting in a dense matrix $C$. It can be expressed as:

$$C_{ij} = \sum_k A_{ik} B_{kj} \tag{1}$$

**Sparse Matrix-Vector Multiplication (SpMV):** The SpMV operation computes the product of a sparse matrix $A$ and a dense vector $x$, resulting in a dense vector $y$. It can be expressed as:

$$y_i = \sum_j A_{ij} x_j \tag{2}$$

| Benchmark | Computation | Comment |
|-----------|-------------|---------|
| SpMM | $A_{ij} = B_{ik}C_{kj}$ | $B$ sparse, $C$ dense |
| SpMV | $y_i = B_{ij}x_j + z_i$ | B sparse, $x, z$ dense |
| SDDMM | $A_{ij} = B_{ij}C_{ik}D_{kj}$ | $B$ sparse, $C, D$ dense |
| MTTKRP | $A_{ij} = \mathcal{X}_{ikl}D_{lj}C_{kj}$ | $\mathcal{X}$ sparse, $C, D$ dense |
| TTV | $A_{ij} = \mathcal{X}_{ijk}z_k$ | $\mathcal{X}$ sparse, $z$ dense |

Table 4: Executed computation by each program in the TACO tasks of CATBench. Each benchmark involves fundamental operations of ML algorithms, such as sparse matrix multiplication or tensor-vector multiplication. Upper-case variables denote matrices, lower-case denote vectors, and curly variables denote tensors. Indices $i$, $j$, $k$, and $l$ are iterated through according to the size of the matrix.

| Benchmark | Computation | Comment |
|-----------|-------------|---------|
| Asum | $\sum_{i=1}^{n} |x_i|$ | $x$ dense |
| GEMM | $C_{ij} = \sum_k C_{ik}D_{kj}$ | $C, D$ dense |
| Scal | $x_i \leftarrow \alpha x_i$ | $x_i$ dense, $\alpha$ scalar |
| Stencil | $A_{i,j} = w_c C_{i,j} + w_n C_{i-1,j} + w_s C_{i+1,j} + w_w C_{i,j-1} + w_e C_{i,j+1}$ | $C$ dense, $w$ 5-tuple |
| Kmeans | $\sum_{i=1}^{K} \sum_{x \in C_i} \|x - \mu_i\|^2$ | See Appendix B.2 |

Table 5: Executed computation by each program in the RISE/ELEVATE tasks of CATBench. Benchmarks involve a combination of elementary operations and prominent ML algorithms. Upper-case variables denote matrices, lower-case denote vectors, and curly variables denote tensors. Indices $i$, $j$, and $k$ are iterated through according to the size of the matrix.

**Sampled Dense-Dense Matrix Multiplication (SDDMM):** The SDDMM operation computes the element-wise product of two dense matrices $C$ and $D$, and then multiplies the result with a sparse matrix $B$. It can be expressed as:

$$A_{ij} = \sum_k B_{ij}C_{ik}D_{kj} \tag{3}$$

**Tensor Times Vector (TTV):** The TTV operation computes the product of a sparse tensor $\mathcal{X}$ and a dense vector $v$ along a specified mode $n$. It can be expressed as:

$$\mathcal{Y}_{i_1...i_{n-1}i_{n+1}...i_N} = \sum_{i_n} \mathcal{X}_{i_1...i_N} v_{i_n} \tag{4}$$

**Matricized Tensor Times Khatri-Rao Product (MTTKRP):** The MTTKRP operation is a key computation in tensor decomposition algorithms, such as CP decomposition. It computes the product of a matricized tensor $X_{(n)}$ and the Khatri-Rao product of factor matrices $A^{(1)}, \ldots, A^{(N)}$ except $A^{(n)}$. It can be expressed as:

$$M_{(n)} = X_{(n)}(A^{(N)} \odot \cdots \odot A^{(n+1)} \odot A^{(n-1)} \odot \cdots \odot A^{(1)}) \tag{5}$$

## B.2 RISE/ELEVATE

The RISE/ELEVATE benchmarks are derived from the RISE/ELEVATE [51] compiler frameworks. These benchmarks cover, dense linear algebra (GEMM, Asum, Scal), clustering (Kmeans), and stencil computations (Stencil). The search spaces involve ordinal parameters, with known and

hidden constraints. The objective is to minimize the execution time of the generated code on CPUs or GPUs.

The RISE/ELEVATE benchmarks in CATBench are derived from the RISE/ELEVATE [51] compiler framework. These benchmarks include:

**Matrix Multiplication (GEMM):** The GEMM benchmark performs dense matrix multiplication of two matrices $A$ and $B$, resulting in a matrix $C$. It can be expressed as:

$$C_{ij} = \sum_k A_{ik} B_{kj} \tag{6}$$

**Asum:** The Asum benchmark computes the sum of absolute values of elements in a vector $x$ of length $n$. It can be expressed as:

$$\text{Asum} = \sum_{i=1}^{n} |x_i| \tag{7}$$

**K-means Clustering:** The K-means benchmark performs K-means clustering on a set of data points $x_1, \ldots, x_n$. The goal is to partition the data points into $K$ clusters, where each data point belongs to the cluster with the nearest mean. The objective function minimized in K-means clustering is:

$$\sum_{i=1}^{K} \sum_{x \in C_i} \|x - \mu_i\|^2 \tag{8}$$

where $C_i$ is the set of data points assigned to cluster $i$, and $\mu_i$ is the mean of the data points in cluster $i$.

**Scal:** The Scal benchmark scales a vector $x$ of length $n$ by a scalar value $\alpha$. It can be expressed as:

$$x_i \leftarrow \alpha x_i \quad \forall i \in \{1, \ldots, n\} \tag{9}$$

**Stencil Computation:** The Stencil benchmark performs a stencil computation on a 2D grid $A$. The value of each cell in the output grid $B$ is computed as a weighted sum of its neighboring cells in the input grid $A$. For a 5-point stencil, it can be expressed as:

$$B_{i,j} = w_c A_{i,j} + w_n A_{i-1,j} + w_s A_{i+1,j} + w_w A_{i,j-1} + w_e A_{i,j+1} \tag{10}$$

where $w_c$, $w_n$, $w_s$, $w_w$, and $w_e$ are the stencil weights for the center, north, south, west, and east neighbors, respectively.

## C  Multi-Objective Results

We additionally visualize the empirical Pareto fronts on the Kmeans, GEMM, and Stencil benchmarks. While runtime and speedup are substantially positively correlated, all benchmarks still provide meaningful trade-offs between the two objectives. Most prominently, Kmeans offers very diverse solutions along the Pareto front, suggesting a complex multi-objective optimization task.

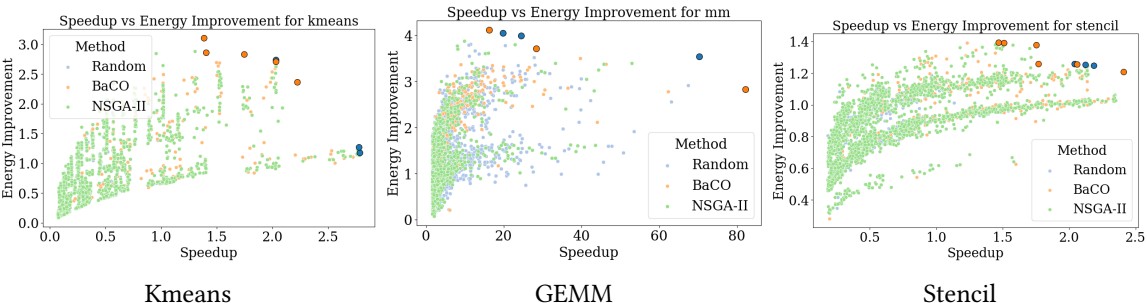

Figure 8: Scatterplot of speedup vs energy efficiency for RISE benchmarks. Pareto fronts are generally diverse, which is most prominently displayed on Kmeans and GEMM. Notably, the speedup factors on GEMM from the median configuration go as high as 80× the runtime of the median.

