# OpenReview forum: "CATBench: A Compiler Autotuning Benchmarking Suite for Black-box Optimization"
_automl.cc/AutoML/2025/ABCD_Track — AutoML 2025 ABCD Track_

### Official Review · Reviewer_Xgkw · 2025-04-26

**Comments To Authors:**

In this paper, the authors introduce CATBench, a comprehensive benchmarking suite designed to evaluate black-box optimization algorithms—especially Bayesian Optimization (BO)—in the context of compiler autotuning. Drawing from realistic use cases across diverse computational domains and hardware architectures, CATBench encompasses a wide array of optimization scenarios tailored to capture the intricate nature of autotuning problems.
The main purpose of the  authors is to address the lack of representative and standardized benchmarks in this field, proposing hence a tool that integrates real-world compiler tasks (e.g., tensor algebra, clustering, image processing). They point out the need of various characteristics such as discrete, categorical, and permutation parameters, known and hidden constraints as well as multi-objective evaluations.

The paper is very well written and organized. The authors propose a state of the art that clearly allow them to highlight their contribution in very active research domain. Their main objectoive is clear. The properties of the proposed benchmarking suite is analyzed in an experimental section, reviewing different interesting characteristics.

To my opinion, this is clearly an interesting and useful contribution, made available for the community.

Pro :

Through its standardized structure and unified interface, CATBench fosters reproducibility, facilitates equitable algorithm benchmarking, and accelerates the development of more robust and scalable optimization strategies.

The benchmarks span multiple compilers (TACO, RISE/ELEVATE) and hardware platforms (CPUs, GPUs), and are provided in a containerized, Python-accessible format with support for surrogate models and scalable deployment.

CATBench encompasses a wide array of optimization scenarios tailored to capture the intricate nature of autotuning problems.

CATBench challenges BO algorithms with complex search spaces—including categorical, ordinal, and permutation-based parameters—as well as with multi-objective trade-offs and opportunities for transfer learning.

Minor remark : the citing style for references must be unified

**Review Confidence:**

4

**Review Rating:**

8

---

### Review · Reproducibility_Reviewer_GAyX · 2025-04-30

**Comments To Authors:**

I thank the authors for submitting their artifact for review. As this work introduces a benchmark, it is of great value and has the potential to have a significant impact on the community. However, my review is based on the consideration of the current version of CATBench's GitHub repository as the supporting artifact of the paper.

**To summarize, I was able to run some minimal examples based on my understanding of the paper and the code in `basic_example.py`, so I can conclude that the benchmark is usable.** Replicating or reproducing the results of experiments presented in the paper, however, requires moderate to major effort (not considering the specific hardware requirements, which are of course not relevant to the review). **However, again, based on my understanding of the code, I guess that those experiments, once run, can be replicated**.

I have some comments and suggestions listed below, which I hope the authors may find helpful, and address to some extent in order to increase the reproducibility and usability of the artifact.

- The repository lacks explicit setup instructions. Although it might be trivial to install a GitHub repo for an experienced user, specific installation instructions can always help and make things easier. These instructions can include information that is not presented in the `requirements.txt` file, for example, the required or recommended/tested Python or Docker versions. For reference, I tested catbench with Python `3.10.12`.
- The repo/artifact can greatly benefit from **specific scripts to run experiments corresponding to each figure, table, or sets of results presented in the paper**. With this, it would also be nice to add code or, e.g., Jupyter notebooks for visualizing or recreating the plots included in the paper (I personally consider such scripts to be a requirement for assessing the reproducibility of an artifact or paper).
- Not considering the information in `README.md`, I could not find any documentation in the repo.
- Some references are not in the same style (see, e.g., the second paragraph of the Introduction section).
- From reading the paper it is unclear to me what the unit of "compute time" in the results section is.
- The (optional) reproducibility checklist is missing.

**Review Confidence:**

4

**Review Rating:**

6

---

### Official Review · Reviewer_TDPD · 2025-05-01

**Comments To Authors:**

The manuscript introduces CATBench, a tool for benchmarking Black-Box optimizers in the context of compiler parameter selection, otherwise known as autotuning. The benchmark is introduced as a complete solution, which integrates a Python interface, a containerized client-server architecture, and a set of test problems derived from the Tensor Algebra Compiler (TACO) and the RISE/ELEVATE tests. To demonstrate the usability of the tool, the authors tested five hardware architectures, three for TACO and two for RISe, and three Black-Box optimization algorithms, i.e., Bayesian Compiler Optimisation (BaCO), NSGA-II and Random Search, with BaCO showing stronger performance out of the three. Negligible differences between hardware architectures were observed. The trade-offs between speed-ups and energy consumption were also examined, showing a strong correlation, which matches our expectations. Multi-fidelity evaluation is possible by changing the complexity of the compilation process, i.e., the number of repeats, the wait time between and after them, and the clearing of the cache. However, this aspect is not elaborated further in the manuscript. The paper is mostly clear for somebody with a limited understanding of compiler tuning. Therefore, I have only minor comments regarding structure and clarity:

1. There is something wrong with the citation format, with some appearing as numbers and others as [Last Name].
2. Section 2.3 - Bayesian Optimization over Discrete Search Spaces: While SMAC is mentioned by name, citations [38] and [8] and their trails are not. Do these refer to specific methods?
3. TACO is first mentioned in Section 2.3 - Benchmarking Suites. The acronym is not introduced then but in Section 3.2
4. Section 3: "Autotune presents unique challenges ... the optimization challenging" This phrase is unusual.
5. Section 3.2.1: Can the authors provide an example of a permutation parameter?
6. Was there any evaluation of the multi-fidelity parameters? What were the defaults?
7. The order of the results was confusing, as the Figures are referenced in different sections and out of order.
8. What were the parameters of BaCO and NSGA-II? The latter did quite poorly, not obtaining any solution in the Pareto front (Figure 6) and performing equal to or worse than the random search (Figure 2,3). Can the authors provide some reasoning for this?
9. Consider using the same scale for Figure 5. No mention is made of gs0 and gs1 in the caption, which is the opposite for omp_num_threads. Is there a reason?

**Review Confidence:**

3

**Review Rating:**

8

---

### Official Review · Reviewer_SM8i · 2025-05-02

**Comments To Authors:**

This paper provides an important resource for the black-box optimization and compiler autotuning communities. CATBench raises the standard for realism and complexity in optimization benchmarks and is poised to support a new wave of algorithmic research. Its real-world tasks, support for mixed parameter types, constraints, and multi-objective/fidelity evaluations make it a significant contribution to the field.

The paper convincingly argues that compiler autotuning presents challenges not well-addressed by standard BO benchmarks, including permutation-based parameters, hidden constraints, and multi-fidelity objectives. It fills a critical gap in the benchmarking ecosystem. However, it has some minor flaws:
1- While CATBench includes hidden and known constraints, it is not fully clear how these are handled or penalized during optimization. Are invalid configurations discarded or modeled? Please add that to your paper
2- Performance comparisons lack statistical analysis (e.g., Wilcoxon tests or confidence intervals) to assess whether observed differences are significant.
3- For further evaluation, it would be better to include additional BO methods for a better comparison study.

I recommend this paper to be accepted for AutoML conference because it proposes an interesting work and brings a high-quality contribution to the field.

**Review Confidence:**

4

**Review Rating:**

9

---

### Meta-Review · Area_Chair_6QhV · 2025-05-09

**Recommendation:** Accept
**Confidence:** 5

**Metareview:**

The paper presents a benchmarking tool for Bayesian optimisation in compiler autotuning. It provides insights into the specific requirements of the domain problem not addressed in other benchmarks and encompasses realistic optimisation scenarios. It is also well-written.
All authors acknowledge the merits of the paper and provide pointers to improve the final draft. The Reproducibility Reviewer GAyX gives clear directions that authors should take into account to improve their code. The paper will impact the research focused on autotuning compilers; hence, I recommend accepting the paper.